# Assessment of Real-Time Quaking-Induced Conversion (RT-QuIC) Assay, Immunohistochemistry and ELISA for Detection of Chronic Wasting Disease under Field Conditions in White-Tailed Deer: A Bayesian Approach

**DOI:** 10.3390/pathogens11050489

**Published:** 2022-04-20

**Authors:** Catalina Picasso-Risso, Marc D. Schwabenlander, Gage Rowden, Michelle Carstensen, Jason C. Bartz, Peter A. Larsen, Tiffany M. Wolf

**Affiliations:** 1Department of Veterinary Population Medicine, University of Minnesota, Saint Paul, MN 55108, USA; picas001@umn.edu; 2Minnesota Center for Prion Research and Outreach, University of Minnesota, Saint Paul, MN 55108, USA; schwa239@umn.edu (M.D.S.); rowde002@umn.edu (G.R.); jbartz@creighton.edu (J.C.B.); plarsen@umn.edu (P.A.L.); 3Facultad de Veterinaria, Universidad de la Republica, Montevideo 11200, Uruguay; 4Department of Veterinary and Biomedical Sciences, College of Veterinary Medicine, University of Minnesota, Saint Paul, MN 55108, USA; 5Minnesota Department of Natural Resources, 5463 West Broadway, Forest Lake, MN 55025, USA; michelle.carstensen@state.mn.us; 6Department of Medical Microbiology and Immunology, School of Medicine, Creighton University, Omaha, NE 68178, USA

**Keywords:** latent class analysis, diagnosis, sensitivity, specificity, real-time quaking-induced conversion, gold standard, immunohistochemistry, ELISA, immunoassays, prion

## Abstract

Chronic wasting disease (CWD) is a transmissible prion disease of the *cervidae* family. ELISA and IHC tests performed postmortem on the medial retropharyngeal lymph nodes (RPLN) or obex are considered diagnostic gold standards for prion detection. However, differences in CWD transmission, stage of infection, pathogenesis, and strain can limit performance. To overcome these uncertainties, we used Bayesian statistics to assess the accuracy of RT-QuIC, an increasingly used prion amplification assay, to diagnose CWD on tonsil (TLN), parotid (PLN) and submandibular lymph nodes (SMLN), and ELISA/IHC on RPLN of white-tailed deer (WTD) sampled from Minnesota. Dichotomous RT-QuIC and ELISA/IHC results from wild (*n* = 61) and captive (*n* = 46) WTD were analyzed with two-dependent-test, one-population models. RT-QuIC performed on TLN and SMLN of the wild WTD population had similar sensitivity (median range (MR): 92.2–95.1) to ELISA/IHC on RPLN (MR: 91.1–92.3). Slightly lower (4–7%) sensitivity estimates were obtained from farmed animal and PLN models. RT-QuIC specificity estimates were high (MR: 94.5–98.5%) and similar to ELISA/IHC estimates (MR: 95.7–97.6%) in all models. This study offers new insights on RT-QuIC and ELISA/IHC performance at the population level and under field conditions, an important step in CWD diagnosis and management.

## 1. Introduction

Chronic Wasting Disease (CWD) is an incurable, progressive, and fatal prion disease affecting cervids worldwide. Since first reported in the 1960s in the U.S. [1,2], CWD has spread to 29 U.S. states and other countries such as Canada, South Korea, Norway, Finland, and Sweden [3,4,5]. As a One Health issue, its impact affects three levels: the health of the animals infected (free-living and captive) [6], ecosystem imbalance due to environmental contamination and cervid population decline [7,8,9], and the potential effects on human health (i.e., still inconclusive zoonotic potential) [10,11,12,13,14,15,16,17,18] and well-being (e.g., hunting heritage, business impacts) [19]. CWD spread and transmission will be based on prion shedding by infected animals and its persistence in the environment to effectively infect new hosts [9,16,18,20,21,22,23,24,25,26,27]. The control and eradication of CWD relies on the early detection of infected animals and their removal from the population to avoid transmission and extensive environmental contamination [1,7,28]. Despite continuous efforts to control CWD globally, success is hampered by limitations of the diagnostic methods used for the detection of infected animals.

The accuracy of a diagnostic test is often estimated relative to a reference test, known as the gold standard. The “gold standard” assumes perfect accuracy in sensitivity, in which infected animals will always be detected by the test, and specificity, in which uninfected animals will never test positive. However, under natural conditions in which the infected population is a heterogeneous mix of animals at various stages of infection or disease progression and true disease status is unknown, disease status (i.e., infected or uninfected) misclassifications can occur, even when using the presumed gold standard approaches to disease detection [29,30]. Currently, immunoassays such as the enzyme-linked immunosorbent assay (ELISA) and immunohistochemistry (IHC) are considered gold standard tests for the detection of CWD when performed on obex and medial retropharyngeal lymph nodes (RPLN). A few studies have estimated the diagnostic performance of ELISA and/or IHC [29,31,32], but they have limited ability to address optimal external field validity. For example, while experimental studies can offer insights on test performance at different stages of disease [29], we often do not know the stage of disease of animals tested in a natural setting. Test accuracy of one assay (ELISA or IHC) has also been assessed relative to the other, with high correlation of prion detection and nondetection at similar stages of infection [32,33,34]. When test performance has been assessed outside of experimental settings, it has been done using samples from multiple populations and across cervid species with limited samples sizes in some cases (e.g., 3 samples from white-tailed deer (WTD)) [31], resulting in much uncertainty around estimates of performance. Thus, true diagnostic accuracy under field conditions (i.e., in which natural infection occurs) remains unknown. Further, studies evaluating amplification-based methods for prion detection have demonstrated that the sensitivity of ELISA and IHC are not 100% [32,34,35,36]. While immunoassays are highly effective in detecting infected animals, particularly those in later or clinical stages of infection [1,37,38], test accuracy can be limited in earlier stages of infection when prion accumulation is low and dissemination in tissues varies based on the route of exposure or prion strain [29,33,39]. Given these limitations, when it comes to evaluating the performance and accuracy of new diagnostic tests for prion detection under natural conditions of transmission, caution should be taken when making “gold standard” assumptions.

The need for a faster, accurate, reliable, cost-effective assay, with the potential for application in antemortem diagnosis, has led to the development of techniques relying on prion amplification, such as the real-time quaking-induced conversion (RT-QuIC) [40]. RT-QuIC has a similar level of accuracy to IHC and ELISA [32,41,42], though it can detect prion-causing CWD (PrPSc) early in time from onset of infection [43]. Yet, like ELISA and IHC, detection may be impacted by the distribution of prions in tissues and sampling method [36,42]. Although RT-QuIC has been proven useful for the detection of infected samples [36,44,45], similar to ELISA/IHC, the estimation of its accuracy at the field level where stage of disease among tested animals is variable and unknown is uncertain.

Given the aforementioned challenges, knowledge of diagnostic test accuracy, as they are applied under conditions of natural transmission, is critically needed by researchers, regulatory agencies, and policy makers. A Bayesian latent class analysis (BLCA) is a useful alternative approach to the assessment of diagnostic test performance that can overcome the constraints related to reliance on a “gold standard” comparison [46,47,48,49]. Briefly, this methodology combines prior literature and expert knowledge on the expected performance of the diagnostic tests and the population prevalence of the disease. In doing so, inherent levels of uncertainty are explicitly modeled with empirical diagnostic data to yield estimates of the diagnostic tests’ sensitivity and specificity with high precision [46].

In this study, we aimed to evaluate the performance of RT-QuIC and ELISA/IHC for the detection of PrPSc in lymphoid tissues from white-tailed deer under field conditions. To do so, samples were collected from wild and captive white-tailed deer in Minnesota to a) quantify the concordance between RT-QuIC and the ELISA/IHC across different sample matrices, and b) estimate the sensitivity and specificity of RT-QuIC and ELISA/IHC using BLCA as compared to the traditional “gold standard” approach. Given its demonstrated capability for earlier prion detection in tissues of infected animals [43], we hypothesized that RT-QuIC would perform as well or better than ELISA/IHC for CWD detection under conditions of natural transmission.

## 2. Results

RPLN, parotid lymph nodes (PLN), palatine tonsils (TLN), and submandibular lymph node (SMLN) tissue samples were obtained from culled, wild (*n* = 61) white-tailed deer (WTD) in southeast Minnesota where CWD infection is persisting in approximately 1% of the WTD population (herein referred to as the “wild population”), and a depopulated CWD + herd of captive WTD (*n* = 46) in the same region (herein referred to as the “captive population”).

### 2.1. Descriptive

As an initial step in evaluating the dichotomous results for CWD diagnosis across tissues and tests, we created contingency tables comparing results from ELISA/IHC performed on RPLN, and those results from RT-QuIC performed on PLN and TLN in the captive population, and on PLN, TLN, and SMLN from the wild population. Results obtained from RT-QuIC were cross-tabulated considering two classification approaches: the standard, in which inconclusive results were computed as negatives, and the inclusive, in which inconclusive results were considered positives (Table 1). The inclusive approach to result classification identified three more animals from each population (*n* = 6) as PrPSc-positive than the standard approach. These included one PLN- and two TLN-positive animals from the wild, and two PLN- and one TLN-positive captive animals. When the RT-QuIC assay was performed on SMLN from the wild population, there were no inconclusive results to assess the different classification approaches.

### 2.2. Test Agreement and Analytical Accuracy Using a “Gold Standard” Approach

We examined the level of agreement between RT-QuIC by both classification approaches and the various tissue types and ELISA/IHC of the RPLN. To do this, we assessed the concordance among different diagnostic test combinations beyond chance using kappa statistics (κ). Agreement observed between RT-QuIC using different tissue matrices ranged between good and excellent (κ: 0.69–0.95), with the least agreement observed between RT-QuIC performed on PLN vs. TLN (Table 2). The concordance between RT-QuIC and ELISA/IHC assays were excellent (κ: 0.91–0.95) or very good (κ: 0.74–0.89), depending on the lymphoid tissue tested and the classification approach applied (Table 2).

For demonstrative purposes, we first estimated the accuracy of RT-QuIC for PrPSc detection using the traditional “gold standard” approach. The performance of RT-QuIC in reference to ELISA/IHC as the gold standard tests of comparison showed a high sensitivity (Se) when using TLN and SMLN (Table 3). RT-QuIC applied to TLN using the inclusive classification approach demonstrated the highest observed Se (0.95, 95% CI: 0.77–0.99). The specificity (Sp) of RT-QuIC diagnosis using any lymphoid tissue and classification scheme was very high (0.96–1.00) (Table 3).

### 2.3. Analytical Accuracy Using a Bayesian Approach (BLCA)

To avoid the “gold standard” assumption, we used a series of two-dependent-test, one-population latent class models to estimate the diagnostic Se and Sp of RT-QuIC (by both result classification approaches), ELISA and IHC, as well as population CWD prevalence. The statistical models combine results from two different diagnostic tests; ELISA or IHC as test one, and RT-QuIC for each tissue sampled respectively as test two, in each population of study independently (wild or captive), with prior information on each parameter to obtain the most probable field values. Prior information informs the most likely range of values for each parameter that are compatible with previously published studies and experts’ opinions while incorporating the knowledge uncertainty (e.g., taking into account sample size, and external validity of those studies). Hence, informative, weakly informative, or non-informative priors were selected based on a lower to a higher level of uncertainty on previous knowledge and experiences.

Latent class models yielded median Se estimates for ELISA/IHC of RPLN between 91 and 94.5% for the wild, and 88.4 and 93.1% for the captive population. Sensitivity of RT-QuIC resulted in higher (92.2–95.1%), similar (92.1%), or lower (88.2–88.3%) median estimates when tested on TLN, SMLN, and PLN, respectively, for the wild population. Lower Se-median estimates were observed when using RT-QuIC in TLN (88.5%) or PLN (73.8–74.3%) in the captive population. The inclusive classification approach for CWD results by RT-QuIC assay resulted in the highest Se (95.14, 95% Posterior Predictive Interval (95% PPI): 76.6–99.8) when applied to TLN in the wild population model, with no differences observed in the estimates generated by standard vs. inclusive criteria in the rest of the latent class models using other tissues or the captive population (Table 4). Specificity median estimates for RT-QuIC (range: 95.5–98.6%) and ELISA/IHC (range: 96.2–97.6%) were similar in all models regardless of the tissue type or population tested, with higher estimates (1–3%) for the standard approach (Table 4); while the prevalence of infection was consistently higher (1–3%) in the wild vs. captive population estimates. Note, while these prevalence estimates were expected based on the structure of the wild population sample set (refer to Methods), it does not represent the true infection prevalence of the wild population in this region.

We structured our models assuming a conditional correlation between tests. This means that we assumed that if an infected animal tested positive (or false negative) by one test (e.g., ELISA/IHC), a similar result would likely be observed by the other test (RT-QuIC) (i.e., conditional correlation among infected animals), and a similar scenario will be observed for uninfected animals (i.e., conditional correlation among non-infected animals). This was primarily based on the fact that each test is based on the detection of the infectious prion in tissue (vs., for example, the detection of an immunological response generated by the animal). However, low conditional correlation estimates were observed between RT-QuIC and ELISA/IHC among infected and non-infected animals, as denoted by the 95% PPI including zero for all models (Table 4).

### 2.4. Sensitivity Analysis

We conducted a sensitivity analysis to determine how much our model structure and prior parameter distributions influenced model results. We did this by running a new set of models that included a two-population model and where uninformative priors were reassigned to individual parameters. Specifically, beta distributions around prior median estimates of Se, Sp, and prevalence were replaced with uninformative uniform distributions (i.e., assuming no prior available information). Sensitivity analyses indicated no effect on the median posteriors of the parameters estimated (variations < 10%) when fitting a two-dependent test, two-population model, however, the DIC was higher (>10 points) in the latter model (Appendix A), which indicates less “goodness of fit” and more complex model than the one originally performed. Similarly, results were repeatable (estimate variations < 8%, and similar DICs) when non-informative priors were assumed, or priors for RT-QuIC of TLN were included, representing low reliance on the priors for the model-generated posterior estimates (Appendix A). The effect of using a conditional dependent model on posterior estimates was assessed by running all models (with reference and uninformative priors) assuming conditional independence. Still, no improvement was observed on the DIC of each model when independence among tests was assumed. Convergence was reached in all models as shown by visual stabilization of the Markov chains and the Gelman-RubinˆR statistic (<1.002) for all parameters.

## 3. Discussion

With the development and application of new diagnostic tests for CWD and other prion diseases, there is a critical need to understand the efficacy of their performance in naturally infected populations, where true disease status is unknown. With a recognition of imperfect testing for disease classification, the current study applied Bayesian statistics to determine the accuracy and best use of two diagnostic assays for CWD in free-ranging and captive WTD. Through these analyses, we demonstrate high Se and Sp of RT-QuIC, particularly when used on TLN, and provide the first set of performance estimates for RT-QuIC as well as ELISA/IHC in both wild and captive settings. We employed this approach in association with the frequently used assessments of test performance, including analysis of test agreement and in reference to an assumed “gold standard” of comparison. While the accuracy of RT-QuIC in reference to the ELISA/IHC as the “gold standard” showed similar patterns of performance to those observed with the BLCA, they were characterized by higher uncertainty and limited validity under natural conditions [50]. Indeed, our results show that none of the diagnostic assays examined, including ELISA/IHC, demonstrate perfect Se or Sp in a naturally infected population. Collectively, our results help to document the utility of the BLCA approach for the assessment of diagnostic test performance in the natural setting [1,46,50,51].

### 3.1. Characteristics of the Tests

Our estimates of ELISA/IHC test performance are the first quantitative estimates of the accuracy of these tests for PrPSc detection in a natural setting. Although the limitations of these tests for diagnosis in cervid populations have been described [7,8,9,11,12,13], the high number of infected animals that could be misclassified as negative (5–10%), based on our ELISA/IHC median sensitivity estimates (Table 4), raises concerns for stakeholders. For example, although the zoonotic potential of CWD is inconclusive and these tests have not been validated as food safety tests, hunters’ use of “not detected” test results in their harvest consumption decisions may be problematic. These findings are also relevant for management agencies using these tests to detect disease and adapt harvest regulations for CWD control. Our findings suggest the importance of the explicit incorporation of these estimates of test performance into the design of population surveillance systems, particularly where there is a need to demonstrate freedom of disease [52].

RT-QuIC exhibited the highest level of Se when performed on TLN and applying an inclusive criterion of classification. Currently, there is a lack of standardization among the research and diagnostic community on the assignment of the dichotomous classification of CWD disease status (positive vs. non-detected) from the amyloid formation process of RT-QuIC. Up to now, a variety of statistical approaches have been employed for disease status classification, such as Mann-Whitney [36,53], Wilcoxon rank test [54,55], or differences in standard deviations of the samples [56,57]. In this study, application of the Mann–Whitney test, as described previously [36], resulted in a small number of samples (*n* = 5, Table 1) with RT-QuIC seeding activity, though they did not meet statistical criteria to be classified conclusively as positive. Thus, through this study, we explored the effect of considering those inconclusive samples as positives (inclusive) or negative (standard). The improved Se observed with TLN samples, even with a small number of inconclusive samples (2/61) that were classified as positive, indicated that this more inclusive classification approach may improve early diagnosis of CWD in WTD, while not significantly impacting Sp.

Diagnosis reached with RT-QuIC using TLN and SMLN had higher than or similar Se and Sp for CWD diagnosis as ELISA/IHC using RPLN. The high sensitivity observed with TLN samples replicated results observed in previous studies of IHC performed on limited (*n* = 5) biopsies of TLN in CWD-infected WTD [58,59], highlighting the importance of this tissue for accurate CWD diagnosis. Additionally, SMLN screened by RT-QuIC had similar diagnostic potential as RPLN (by ELISA/IHC) and TLN (by RT-QuIC), which increases the opportunities for diagnosis when other lymphoid tissues are not available. Given these results and those of previous studies [36,42] that reveal variation in prion detection across lymphoid tissues of infected animals, there may be advantages in sampling and pooling multiple tissue types to improve CWD detection. That was not something examined in this study, but may be an important future direction.

Poor PrPSc detection obtained with the use of PLN may be due to the inclusion of samples from animals in earlier stages of infection (<4 months post-exposure) with oropharyngeal prion entry [43,60], or the limited involvement of anatomic subsites drained by this lymph node in CWD pathogenesis [6]. The critical role of parotid glands in the rumination cycle has been associated with an increase in the flow of lymph through the RPLN, which can potentially explain the delay in accumulation of PrPSc in PLN and higher detection in RPLN [37,61].

Consistently higher posterior RT-QuIC Se estimates of the wild vs. captive population indicate different levels of detection by the same diagnostic test according to the population under study. This finding suggests that these two populations may vary in their proportions of animals at certain stages of the disease, PrPSc strains, or predominant routes of prion transmission. Certainly, the route of transmission and time post-infection can influence the prion load in the lymphoid tissues under study, and that can impact the accuracy of tests for identifying infected animals [33,34]. In addition, the PrPSc strains infecting each population may differ, with differing levels of pathogenicity that could affect the progression and dissemination of prions throughout the examined lymphoid tissues [62,63]. This trend was not observed with ELISA/IHC diagnostic results, which may suggest that prion accumulation in the RPLN, regardless of these possible epidemiologic (e.g., route, dose, age of infection) and strain variations, may be a consistent feature of the CWD PrPSc.

### 3.2. Model Assumptions

To estimate the sensitivity and specificity of RT-QuIC and ELISA/IHC, we fitted models using one population to ease interpretation and because, based on DIC statistics, a seven-parameter model (Se_1_, Se_2_, Sp_1_, Sp_2_, rhoD, rhoDc, Prev) was preferred over an eight-parameter one (Se_1_, Se_2_, Sp_1_, Sp_2_, rhoD, rhoDc, Prev_1_, Prev_2_). The low correlation observed between tests results, although not initially expected, could be due to different levels of PrPSc concentration across the lymphoid tissues used with each test type, impacting disease detection [37].

We employed weakly informative priors (i.e., distributions with higher uncertainty) for the Se and Sp estimates for ELISA/IHC because of the limited studies of performance available, of which the majority were applied under laboratory conditions or with limited sample size [29,31,32,64]. We assumed that ELISA and IHC accuracy was similar given the previous review available, indicating that both diagnostic assays performed similarly [45]. Despite these limitations, Se analysis demonstrated that these a priori assumptions had no effect on the results.

Priors established for the population prevalence of the wild WTD models were higher than those reported for the southeast Minnesota WTD wild population, and more similar to those observed in depopulated captive populations in the state [65]. Indeed, CWD prevalence in WTD in southeastern Minnesota has been estimated at approximately 1% through hunter-harvest surveillance [66], and prevalence among depopulated captive herds have ranged widely between 8 and 100%, depending on herd size [67]. However, we expected a higher prevalence of CWD-positive animals in the wild WTD data since RT-QuIC testing was conducted on culled animals suspected of having disease in areas of previous detections and not randomly sampled. In addition, a subset of these data that were used in this study were enriched with positive samples for RT-QuIC validation purposes [36].

## 4. Materials and Methods

Guidelines from the STARD-BLCM were followed to describe the materials and methods in our study [68].

### 4.1. Source Populations

Samples of lymphoid tissue were collected in 2019 (wild population) and 2021 (captive population) from two distinct WTD populations in Minnesota (MN), with a cross-sectional study design.

Lymphoid tissues were sampled from 46 culled captive asymptomatic animals originating from a CWD outbreak farm previously detected in southeastern MN. The farm was confirmed CWD-positive by the Minnesota Board of Animal Health (BAH) and United States Department of Agriculture (USDA); tissues were collected at the time of depopulation for the control of CWD [66]. Furthermore, lymphoid tissue collected from 61 wild WTD were selected from a pool of asymptomatic animals culled for CWD control in Fillmore and Winona counties in southeastern Minnesota. CWD infection is persisting in the wild WTD population in these counties, with an estimated prevalence of approximately 1% (based on ELISA/IHC results), and has been managed for its control by the Minnesota Department of Natural Resources (MNDNR) for more than five and three years, respectively [66]. The 61 animals were selected from a pool of 500 by MNDNR collaborators with a goal of enriching the sample set with ELISA/IHC positive, as well as -negative samples for the purpose of blind interrogation of disease status by RT-QuIC, as previously described [36].

### 4.2. Samples and Diagnostic Assays

Official authorities (MNBAH and MNDNR) collected RPLN for CWD detection in both populations of WTD. Bilateral RPLN collected from captive WTD were diagnosed by IHC at the National Veterinary Services Laboratory (Ames, IA, USA). Bilateral RPLN of wild WTD were submitted for diagnosis at the Colorado State University Veterinary Diagnostic Laboratory (CSU VDL, Fort Collins, CO, USA) for screening by ELISA, using the Bio-Rad TeSeE Short Assay Protocol (SAP) Combo Kit (BioRad Laboratories Inc., Hercules, CA, USA). For each animal, a homogenate was produced using three subsamples from each RPLN and tested as a pooled sample of both RPLN subsamples. ELISA-positive samples were confirmed by IHC, as described [30]. Samples with an ELISA Optical Density (O.D.) > 0.100 and/or IHC positive results were classified as CWD-positive, otherwise, they were classified as CWD-nondetected [64]. Results from the diagnostic tests performed were provided and included in this study.

Bilateral lymphoid tissue from the palatine tonsils (TLN), parotid (PLN), and submandibular lymph nodes (SMLN) (the latter only for the wild population) were sampled, pooled into a single homogenate, and screened by RT-QuIC by the Minnesota Center for Prion Research and Outreach (MNPRO) laboratories using a blinded approach. RPLN were not available from either population for RT-QuIC analysis as they were used for the regulatory testing (IHC and/or ELISA). The sample collection and RT-QuIC methodology have been described previously [36]. Briefly, the RT-QuIC methods are as follows. Bilateral samples were dissected with disposable tools on fresh disposable benchtop paper. A 10% (*w*:*v*) suspension was made by adding 100 mg of tissue to 900 µL of phosphate-buffered saline (PBS). Homogenized samples were diluted further to 10^−3^ in dilution buffer (0.1% sodium dodecyl sulfate, 1 × PBS, N-2 Supplement (Life Technologies Corporation, Carlsbad, CA, USA)), and 2 µL were added to 98 µL of RT-QuIC master mix (1 × PBS, 500 µM EDTA, 50 µM Thioflavin T, 300 mM NaCl, and 0.1 mg/mL Syrian hamster rPrP prepared in house [36]) into each well on a black 96-well plate with clear bottoms. Plates contained appropriate controls and were sealed with clear tape then shaken on a BMG FLUOstar^®^ Omega microplate reader (BMG LABTECH Inc., Cary, NC, USA) at 700 rpm, double orbital for 57 s, then rested for 83 s, repeated 21 times, then the fluorescence was recorded. The temperature was set at 42 °C and the cycles were repeated over 46 h.

RT-QuIC results were dichotomized using two criteria, standard and inclusive. The standard approach classified as positive those samples in which at least one of the sample quadruplicates tested showed an amyloid formation as relative fluorescent units in a sample over time with significantly (Mann-Whitney *p*-value ≤ 0.05) higher intensity than the negative control. Analyses were performed using GraphPad Prism software (version 9, GraphPad Software, San Diego, CA, USA), as described previously [36]. Because samples with low quantities of prion may result in statistically insignificant rates of amyloid formation yet still contain prion [69], we also evaluated test performance under an alternate classification approach (inclusive) where such samples that might be considered “suspect” for CWD prion are classified as positive. Thus, the inclusive approach classified as positive all those with any trace of amyloid formation, regardless of the statistical significance.

### 4.3. Statistical Models

Agreement among dichotomous results using both classification approaches obtained with RT-QuIC from TLN, PLN, SMLN (when applicable), and ELISA/IHC from RPLN were estimated using kappa statistics [70] applied in R software v4.1.0 [71].

Analytical sensitivity and specificity of RT-QuIC in different lymphoid tissues were initially assessed using ELISA/IHC in RPLN as a gold standard diagnostic reference and calculated using Epitools Epidemiological Calculators [72].

Two-dependent test, one-population Bayesian latent class models were applied in the absence of a gold standard to estimate diagnostic Se, and Sp of RT-QuIC (using standard and inclusive classification approaches), as test one, and ELISA in the wild or IHC in the captive WTD population, as test two in each model, respectively [46,73]. The two tests used in each model were assumed to have conditional correlated Se and Sp (i.e., dependent tests), as described previously [74]. We considered this an appropriate assumption, because all tests relied on the detection of prion within tissues extracted from the same region of the body for diagnosis. In other words, both diagnostic tests were targeting the antigen (PrPSc), hence the result of one test is likely to be correlated with the other [73]. We selected a one-population model, because each population (i.e., wild and farmed) was assumed to have a unique prevalence since the behavior and/or management of the two populations is very distinct and likely affects disease transmission among individuals [75].

The Se and Sp for the ELISA/IHC were informed by priors that followed the beta-distribution, built based on previous literature [29,31,32]. The beta priors for the two-population prevalence were built based on previous reports of culled CWD-infected herds from the MNBAH, and opinions elicited from experts working on CWD in Minnesota in the field, laboratory, and epidemiology (Table 5). Note that the bounds on the prevalence estimate for the wild WTD population was set higher than would be expected for the true prevalence in southeastern Minnesota due to the approach used to assemble the data set used in this experiment, where CWD-positive samples were enriched in the sample set to ensure a sufficient number of positive and negative samples for analysis. Beta distributions were fitted using the software Beta buster (version 1.0, Center for Animal Disease Modeling and Surveillance, University of California, Davis. Davis, CA, USA). Limited prior knowledge on the performance of RT-QuIC was available, with most studies performed on RPLN [42,53,56,64,76]. While some studies were performed on TLN samples in WTD [43,58,77], no reports on PLN or SMLN were obtained. Given that we did not assess the performance of RT-QuIC on RPLN, and because the small sample sizes used for TLN did not provide robust information for building priors, we opted to use non-informative priors that followed the uniform distribution (0.1) to inform the Se and Sp estimates for the RT-QuIC assay on all lymphoid tissues.

Models were fitted and run using OpenBUGS 3.2.2 [78] via the R2OpenBUGS package [79] from the R software v4.1.0 [71]. Posterior estimates were obtained after 5000 iterations of the model and the elimination of early samples (500) as a burn-in period. One of every 10 samples was selected to eliminate potential autocorrelation (i.e., thinning). Model convergence was assessed with the Gelman-Rubin^R statistics and visually by observing the stabilization of plots resulting from three Markov chain Monte Carlo runs [80].

### 4.4. Sensitivity Analysis

A sensitivity analysis was conducted to assess the impact of model structure and selected priors on the posterior estimates. For this analysis, we compared the variation of posterior median and probability intervals (PPI) between the initial models and (a) two-dependent test, two-populations models with similar priors, and (b) two-dependent test, one-population model where non-informative priors (e.g., uniform [0,1]) were substituted sequentially for the beta priors of each parameter under study (i.e., Se_2,_ Sp_2_, Prev_1_, Prev_2_). We had one exception to the latter approach where the model assessing the performance of RT-QuIC in TLN, where we substituted the uninformative uniform priors with weakly informative (i.e., high uncertainty) beta priors informed by the limited number of studies from the literature (Table 5) [43,58,77]. The fit of the models was evaluated using the deviance information criterion (DIC) [81].

## 5. Conclusions

To conclude, our robust results, supported by model checking and sensitivity analyses, revealed high diagnostic Se and Sp of RT-QuIC for PrPSc detection in TLN and SMLN under field conditions. We also show that RT-QuIC has improved performance over ELISA/IHC when the approach to dichotomous result classification is relaxed, an approach that can enhance Se without compromising Sp. Among the tissue types tested by RT-QuIC, the use of TLN and SMLN perform equivalently to ELISA/IHC of RPLN, with TLN demonstrating the highest Se of detection. Based on these results, we recommend avoiding the use of PLN for CWD diagnosis in WTD due to lower Se of detection. Our results build on the understanding of the performance of RT-QuIC and ELISA/IHC at the population level and under natural conditions, which represents an important step forward with respect to diagnostic tests for the management of CWD.

## Figures and Tables

**Table 1 pathogens-11-00489-t001:** Cross-tabulation of results obtained for different matrices and classification approaches (cutoff) with RT-QuIC and the ELISA/IHC tests for sampled wild and farmed populations of white-tailed deer in Minnesota.

RT-QuIC			ELISA/IHC RPLN		
Sample	Cutoff	Result	Negative	Positive	Overall
Parotid lymph node	Standard	Negative	47	2	49
		Positive	1	11	12
	Inclusive	Negative	46	2	48
		Positive	2	11	13
Palatine Tonsil	Standard	Negative	47	1	48
		Positive	1	12	13
	Inclusive	Negative	46	0	46
		Positive	2	13	15
Submandibular lymph node	Standard	Negative	48	1	49
	Inclusive ^a^	Positive	0	12	12
TOTAL WILD			48	13	61
Parotid lymph node	Standard	Negative	37	3	40
		Positive	0	6	6
	Inclusive	Negative	35	3	38
		Positive	2	6	8
Palatine Tonsil	Standard	Negative	37	1	38
		Positive	0	8	8
	Inclusive	Negative	36	1	37
		Positive	1	8	9
TOTAL CAPTIVE			37	9	46

Standard: RT-QuIC results computing inconclusive samples as negatives; Inclusive: RT-QuIC results computing inconclusive samples as positives; ELISA: enzyme-linked immunosorbent assay; IHC: immunohistochemistry; RPLN: medial retropharyngeal lymph node. ^a^ Results were similar applying standard and inclusive result classification approaches.

**Table 2 pathogens-11-00489-t002:** Kappa statistic (κ) for agreement (left-lower, colored corner) and the 95% Confidence Interval (right-upper, uncolored corner) between all combinations of diagnostic tests performed on samples from wild and captive white-tailed deer in Minnesota. Colors shown in the left-lower corner indicate the degree of concordance observed. Good agreement in blue (κ = 0.6–0.8), very good in yellow (κ = 0.8–0.9), and excellent in green (κ > 0.9). The bold font indicates the confidence interval for excellent concordance.

		RT-QuICsd		RT-QuICin		RT-QuIC	ELISA/IHC
		TLN	PLN	TLN	PLN	SMLN	RPLN
RT-QuICsd	TLN		0.71–0.98	**0.82–1.00**	0.61–0.92	**0.85–1.00**	**0.82–1.00**
RT-QuICsd	PLN	0.84		0.61–0.92	**0.80–1.00**	**0.76–1.00**	0.67–0.96
RT-QuICin	TLN	0.92	0.76		0.52–0.86	0.70–1.00	0.78–1.00
RT-QuICin	PLN	0.76	0.91	0.69		0.68–1.00	0.58–0.90
RT-QuIC	SMLN	0.95	0.9	0.86	0.85		**0.85–1.00**
ELISA/IHC	RPLN	0.91	0.82	0.89	0.74	0.95	

PLN: parotid lymph nodes; RPLN: medial retropharyngeal lymph nodes; TLN: palatine tonsils; SMLN: submandibular lymph nodes. RT-QuIC: real-time quaking-induced conversion; RT-QuICsd: real-time quaking-induced conversion using the standard cutoff; RT-QuICin: real-time quaking-induced conversion using the inclusive cutoff; ELISA: enzyme-linked immunosorbent assay; IHC: immunohistochemistry.

**Table 3 pathogens-11-00489-t003:** Analytical sensitivity (Se) and specificity (Sp), and 95% Confidence Intervals (95% CI) of RT-QuIC of various lymphoid tissues using ELISA/IHC of medial retropharyngeal lymph nodes as a “gold standard” of comparison.

	Cutoff	Se	95% CI	Sp	95% CI	N
RT-QuIC PLN	Standard	0.7727	0.5463–0.9218	0.9882	0.9362–0.9997	107
	Inclusive	0.7727	0.5463–0.9218	0.9529	0.8839–0.9870	
RT-QuIC TLN	Standard	0.9091	0.7084–0.9888	0.9882	0.9362–0.9997	107
	Inclusive	0.9545	0.7716–0.9988	0.9647	0.9003–0.9927	
RT-QuIC SMLN	NA	0.9231	0.6397–0.9981	1.0000	0.926–1.0000	61

Not applicable (NA) indicates when different cutoff approaches were not plausible to assess. Number of animals sampled and tested by each lymphoid tissue (N). PLN: parotid lymph nodes; TLN: palatine tonsils; SMLN: submandibular lymph nodes.

**Table 4 pathogens-11-00489-t004:** Posterior estimates (median and 95% posterior probability interval) for RT-QuIC and ELISA/IHC sensitivities, specificities, correlation terms for infected (rhoInf) and non-infected (rhoNInf) animals, and prevalence distributions obtained for each model applied for naturally infected wild and captive populations.

Pop	Diagnostic Test			Posteriors Estimates				
	Assay	Matrix	DIC	Sensitivity	Specificity	Prevalence	rhoInf	rhoNInf
Wild	RT-QuICsd	PLN	13.9	0.883 (0.618, 0.994)	0.975 (0.902, 0.999)	0.201 (0.116, 0.312)	0.014 (−0.173, 0.351)	0.061 (−0.033, 0.320)
	ELISA	RPLN		0.921 (0.730, 0.994)	0.966 (0.914, 0.993)			
	RT-QuICin	PLN	14.3	0.882 (0.618, 0.994)	0.962 (0.876, 0.998)	0.207 (0.119, 0.319)	0.015 (−0.182, 0.357)	0.038 (−0.044, 0.273)
	ELISA	RPLN		0.904 (0.692, 0.992)	0.966 (0.915, 0.992)			
	RT-QuICsd	TLN	13.9	0.922 (0.693, 0.997)	0.975 (0.901, 0.999)	0.210 (0.123, 0.318)	0.049 (−0.129, 0.396)	0.055 (−0.023, 0.312)
	ELISA	RPLN		0.923 (0.739, 0.994)	0.971 (0.923, 0.994)			
	RT-QuICin	TLN	14.6	0.951 (0.766, 0.998)	0.963 (0.876, 0.971)	0.224 (0.134, 0.337)	0.097 (−0.093, 0.424)	0.034 (−0.035, 0.271)
	ELISA	RPLN		0.910 (0.711, 0.993)	0.976 (0.933, 0.995)			
	RT-QuIC ^+^	SMLN	13.5	0.921 (0.698, 0.997)	0.986 (0.928, 0.999)	0.202 (0.118, 0.309)	0.044 (−0.110, 0.386)	0.087 (−0.018, 0.356)
	ELISA	RPLN		0.945 (0.779, 0.996)	0.971 (0.922, 0.994)			
Captive	RT-QuICsd	PLN	13.0	0.743 (0.384, 0.980)	0.982 (0.906, 0.999)	0.184 (0.085, 0.313)	−0.056 (−0.310, 0.237)	0.079 (−0.032, 0.354)
	IHC	RPLN		0.927 (0.722, 0.995)	0.960 (0.901, 0.991)			
	RT-QuICin	PLN	13.9	0.738 (0.389, 0.980)	0.950 (0.837, 0.997)	0.202 (0.093, 0.343)	−0.075 (−0.350, 0.253)	0.024 (−0.064, 0.264)
	IHC	RPLN		0.884 (0.639, 0.991)	0.961 (0.899, 0.991)			
	RT-QuICsd	TLN	12.8	0.885 (0.591, 0.995)	0.982 (0.907, 0.999)	0.199 (0.103, 0.331)	0.013 (−0.173, 0.358)	0.071 (−0.028, 0.342)
	IHC	RPLN		0.931 (0.733, 0.995)	0.969 (0.917, 0.993)			
	RT-QuICin	TLN	13.1	0.885 (0.583, 0.995)	0.968 (0.873, 0.998)	0.211 (0.111, 0.340)	0.018 (−0.202, 0.364)	0.045 (−0.041, 0.295)
	IHC	RPLN		0.907 (0.690, 0.992)	0.968 (0.916, 0.993)			

RT-QuICsd: real-time quaking-induced conversion using the standard cutoff; RT-QuICin: real-time quaking-induced conversion using the inclusive cutoff; ELISA: enzyme-linked immunosorbent assay; IHC: immunohistochemistry. (^+^) no difference among results from different cutoff approaches PLN: parotid lymph nodes; RPLN: medial retropharyngeal lymph nodes; TLN: palatine tonsils; SMLN: submandibular lymph nodes.

**Table 5 pathogens-11-00489-t005:** Prior estimates (Mode and 5th percentiles) for sensitivity, specificity of the ELISA/IHC tests, and prevalence for the models implemented, and sensitivity analysis.

	Priors Estimates			
	Parameter	Beta distribution	Mode & 5th perc	References
Captive	Prevalence	α: 2.2011, β: 7.8065	15 (>5)	Experts opinion &
Wild	Prevalence	α: 2.132, β: 3.6413	30 (>70)	MNBAH available data
ELISA/IHC	Sensitivity	α: 6.8414, β: 1.3074	95 (>70)	[29,31,32,64]
	Specificity	α: 53.5808, β: 2.6262	97 (>90)	
RT-QuIC (TLN)	Sensitivity	α: 4.607, β: 1.9017	80 (>40)	[43,58,77]

## Data Availability

All data generated for this study are included in the manuscript and/or Appendix A.

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
