# Peer review of "Assessment of Real-Time Quaking-Induced Conversion (RT-QuIC) Assay, Immunohistochemistry and ELISA for Detection of Chronic Wasting Disease under Field Conditions in White-Tailed Deer: A Bayesian Approach"

_pathogens, 2022, doi:10.3390/pathogens11050489_

Round 1
Reviewer 2 Report
The manuscript entitled “Assessment of real-time quaking-induced conversion (RT-QuIC) assay, immunohistochemistry and ELISA for detection of chronic wasting disease under field conditions in white-tailed deer: a Bayesian approach” (1679206) by Picasso-Risso et al. describes a critical comparison of chronic wasting disease detection techniques.
Chronic wasting disease is a highly contagious prion disease in cervids that is spreading throughout many US states and other jurisdictions. Detection of chronic wasting disease infection often relies on techniques that use postmortem tissues from the lymphoid system. However, determining the sensitivity and specificity of these assays is complicated by the fact that even the widely used “gold standard” detection technique, immunohistochemistry, is an imperfect tool.
In the current manuscript, the authors compare a modern amplification assay - real-time quaking-induced conversion (RT-QuIC) - with more traditional assays such as immunohistochemistry and a commercial ELISA-based detection kit. The study provides detailed statistical comparisons in support of RT-QuIC as an assay of similar sensitivity and specificity as the traditional “gold standard” assays. Overall, the manuscript provides a solid evaluation of the RT-QuIC assay and its use on samples from captive and wild deer populations. However, the following points need to be addressed to improve the manuscript:
- The manuscript uses the term “field conditions” in the title and in other places, but does not provide a clear-cut definition of the term and how to distinguish “field conditions” from laboratory conditions.
- The manuscript discusses how “differences in CWD transmission, stage of infection, and pathogenesis can limit performance” of the CWD detection assays. Potential effects of different CWD strains should be included in the discussion, even if the experimental data were obtained with one CWD strain only.
- The text does not mention if the assays were conducted in a blinded manner or if the experimentalists were are of the prior results from the other assays. While only a formality, it would still be useful to include this information.
- The legend to table 2 mentions “RT-QuICsv: real-time quaking-induced conversion using the severe cutoff”. However, this is the only place a “severe cutoff” is mentioned. Please elaborate or remove this term, if it is a leftover from an earlier version of the manuscript.
